

# A systematic review of pediatric clinical trials of high dose vitamin D

Nassr Nama[1,2], Kusum Menon[2], Klevis Iliriani[3], Supichaya Pojsupap[2], Margaret Sampson[4], Katie O'Hearn[2], Linghong (Linda) Zhou[1], Lauralyn McIntyre[5], Dean Fergusson[6] and James D. McNally[1,2]

[1] Faculty of Medicine, University of Ottawa, Ottawa, Ontario, Canada
[2] Department of Pediatrics, Children's Hospital of Eastern Ontario, Ottawa, Ontario, Canada
[3] School of Medicine, Trinity College, Dublin, Ireland
[4] Department of Volunteers, Communication and Information Resources, Children's Hospital of Eastern Ontario, Ottawa, Ontario, Canada
[5] Ottawa Hospital Research Institute, Ottawa, Ontario, Canada
[6] Clinical Epidemiology Program, Ottawa Hospital Research Institute, Ottawa, Ontario, Canada

Corresponding author
James D. McNally,
dmcnally@cheo.on.ca

## ABSTRACT

**Background.** Due to inadequate UV exposure, intake of small quantities of vitamin D is recommended to prevent musculoskeletal disease. Both basic science and observational literature strongly suggest that higher doses may benefit specific populations and have non-musculoskeletal roles. Evaluating the evidence surrounding high dose supplementation can be challenging given a relatively large and growing body of clinical trial evidence spanning time, geography, populations and dosing regimens. Study objectives were to identify and summarize the clinical trial literature, recognize areas with high quality evidence, and develop a resource database that makes the literature more immediately accessible to end users.

**Methods.** Medline (1946 to January 2015), Embase (1974 to January 2015), and Cochrane databases (January 2015), were searched for trials. All pediatric (0–18 years) trials administering doses higher than 400 IU (<1 year) or 600 IU (≥1 year) were included. Data was extracted independently by two of the authors. An online searchable database of trials was developed containing relevant extracted information (http://www.cheori.org/en/pedvitaminddatabaseOverview). Sensitivity and utility were assessed by comparing the trials in the database with those from systematic reviews of vitamin D supplementation including children.

**Results.** A total of 2,579 candidate papers were identified, yielding 169 trials having one or more arms meeting eligibility criteria. The publication rate has increased significantly from 1 per year (1970–1979) to 14 per year (2010–2015). Although 84% of the total trials focused on healthy children or known high risk populations (e.g., renal, prematurity), this proportion has declined in recent years due to the rise in trials evaluating populations and outcomes not directly related to the musculoskeletal actions of vitamin D (27% in 2010s). Beyond healthy children, the only pediatric populations with more than 50 participants from low risk of bias trials evaluating a clinically relevant outcome were prematurity and respiratory illness. Finally, we created and validated the online searchable database using 13 recent systematic reviews. Of the 38 high dose trials identified by the systematic review, 36 (94.7%) could be found within the database. When compared with the search strategy reported in each systematic review, use of the database reduced the number of full papers to assess for eligibility by 85.2% (±13.4%).

**Conclusion.** The pediatric vitamin D field is highly active, with a significant increase in trials evaluating non-classical diseases and outcomes. Despite the large overall number there are few high quality trials of sufficient size to provide answers on clinical efficacy of high-dose vitamin D. An open access online searchable data should assist end users in the rapid and comprehensive identification and evaluation of trials relevant to their population or question of interest.

## INTRODUCTION

Vitamin D is a steroid-based hormone familiar to health care providers, patients and the media. It is well established that appropriate body stores of vitamin D are essential to musculoskeletal health (*Parfitt et al., 1982*; *Beck-Nielsen et al., 2009*; *Melamed & Kumar, 2010*). As reliance solely on sun (UV) exposure results in high rates of vitamin D deficiency (*Robinson et al., 2006*; *Ahmed et al., 2011*; *Merewood et al., 2012*; *Thacher, Fischer & Pettifor, 2014*), multiple scientific agencies have recommended daily supplementation with small quantities of vitamin D (*Ross et al., 2011*). Despite the success of this approach in reducing the incidence of vitamin D deficiency related electrolyte disturbances and rickets, there continues to be significant interest in alternative high dose vitamin D supplementation strategies. Potential explanations include concern that specific pediatric populations remain at risk for vitamin D deficiency despite recommended dosing (*Aguirre Castaneda et al., 2012*) and that higher doses of vitamin D may protect against or improve outcomes for a wide range of non-musculoskeletal diseases involving the immune, respiratory, and cardiovascular systems (*Brehm et al., 2010*; *Levin et al., 2011*; *Gray et al., 2012*; *Abrams, Coss-Bu & Tiosano, 2013*; *McNally et al., 2015*; *Tomaino et al., 2015*; *Cadario et al., 2015*). Presumably due to uncertainty surrounding the benefit of and best approach to vitamin D supplementation, there has been a growing body of pediatric clinical trial literature. This work spans time, geography, populations (disease states), dosing regimens, and outcome measures. These factors, combined with the large number of adult trials, animal studies and observational literature make the available evidence difficult to find, synthesize and translate to clinical practice or cutting edge research. To assist clinicians and researchers we have sought to identify, describe, and quantify the existing clinical trial literature of high-dose vitamin D supplementation in children through the completion of a systematic review.

The objective of this systematic review was to describe the populations, dosing regimens, methodologies and outcome measures and evaluate how they have varied across geography and time. In addition, we sought to determine the areas where there may be sufficient quantity of high quality evidence to evaluate the benefits of high-dose vitamin D on clinically relevant outcomes. Finally, to assist end user groups in the identification of trials

relevant to their specific patient populations, policy development or research areas, we sought to develop an online trial database searchable by keyword and study characteristics.

## MATERIAL AND METHODS

Study protocol and objectives were established a priori (PROSPERO protocol registration number: CRD42015016242) and reported here according to the PRISMA guidelines of systematic reviews (Table S1) (*Moher et al., 2009*).

### Eligibility criteria

Studies were eligible for inclusion in this systematic review if they satisfied all of the following criteria: (1) Uncontrolled, controlled non-randomized, or randomized controlled trial (RCT); (2) the study involved children; and (3) the study administered cholecalciferol (D3) or ergocalciferol (D2) above the Institute of Medicine (IOM) age specific Recommended Dietary Allowance (RDA) or Adequate Intake (AI). The AI (infants) has been set at 400 IU, with RDA set at 600 IU for those older than one year. As adequate dosing in low birth weight and premature neonates is less well defined, trials administering any dose were considered eligible in these populations. Only trials published in English, French, Spanish, or German were included. Studies were excluded if: they administered Vitamin D as part of a formula, or mixed with food, and dosage was not controlled or consistently delivered; there were no patients less than or equal to 15 years of age; or the study group included patients older than 18 years and did not present data separately for children and adults.

### Identification of studies

The search strategy has been previously described (*McNally et al., 2015*). Medline (1946), Embase (1970), and the Cochrane Central Register of Controlled Trials (2005)  were searched in January 2014 and updated in January 2015 using the Ovid interface. No date, language, or study design limits were applied to the electronic search. The grey literature search included a citation review of all eligible articles, and 24 systematic reviews of vitamin D in children (Appendix S1). The Medline search strategy (Appendix S2) was developed by a librarian (MS) and peer reviewed by another (Lorie Kloda, MLIS, PhD), using the PRESS (Peer Review of Electronic Search Strategies) standard (*Sampson et al., 2009*).

Two of the study authors independently reviewed the citations through three sets of screening questions to determine eligibility (Table S2). Level 1 screening was performed using Mendeley (Mendeley Desktop, version 1.13.8), and those citations that could not be excluded were uploaded to DistillerSR (Evidence Partners, Inc., Ottawa, Canada) for the second and third levels of screening where the full text was assessed for eligibility by two authors, with conflicts resolved by a third. A single author determined the eligibility of articles written in German or Spanish. In the situation of a trial having produced multiple publications, we selected the largest or most complete report; if the two reports described different outcomes in the same trial, all assessed outcomes where listed under the largest report.

## Data collection and analysis

Data was extracted from eligible articles and reviewed independently by two of the authors (NN, KI, KO, SP, JDM). Data was collected using the REDCap system (Research Electronic Data Capture) (*Harris et al., 2009*). Any 25-hydroxyvitamin D (25OHD) data reported only in graphs was extracted using DigitizeIt software (http://www.digitizeit.de/, Germany). Study populations were stratified into three groups: (i) Healthy children with the level of 25OHD and/or bone health as a primary outcome, (ii) diseases classically linked to vitamin D or known to affect its pharmacokinetics (prematurity, renal, rickets, malabsorption, epileptic medications) (*Canadian Agency for Drugs and Technologies in Health, 2015*), and (iii) studies involving children with non-classical diseases or targeting non-classical outcomes. This stratification of classical vs. non-classical diseases follows the guidelines established by several endocrine societies. This separation is relevant as testing for 25OHD levels may only be funded (Canada for example) in patients with one of the conditions classically associated with vitamin D deficiency (*Provincial Programs Branch, Goverment of Ontario, 2010*; *Canadian Agency for Drugs and Technologies in Health, 2015*).

Vitamin D dosing regimens were placed into one of three frequency groups (daily, weekly/bi-weekly, and single/intermittent) and 4 dosing groups (<1,000 IU, 1,000–3,999, 4,000–39,999, >40,000 IU) categories. Consistent with our systematic review evaluating change in 25OHD by dosing regimen, results were presented by study arm, as publications frequently evaluated more than one high dose regimen (*McNally et al., 2015*). Where applicable, we also identified whether and how trials varied the dose (e.g., weight, age or body surface area (BSA)) and determined the maximum dose administered based on the description provided for the study participants in the results. Each study was assessed using Cochrane risk of bias tool (*Higgins & Green, 2011*). Areas where there might be sufficient high quality research to address clinical efficacy were determined by cross-referencing low risk of bias studies with population-outcome data and number of participants enrolled.

## Statistical analysis

Data analysis was performed using SAS (version 9.3; SAS Institute, Cary, NC, USA) and GraphPad Prism (version 6.0.5; GraphPad Software, Inc., La Jolla, CA, USA). Figures were generated using SigmaPlot (version 12.3.0.36; Systat Software, Inc., Germany). Chi-square and Fisher's exact tests were used to compare features between different decades and/or regions.

## Online database

Using Knack software, we developed an online database with relevant information extracted from each identified trial (http://www.cheori.org/en/pedvitaminddatabaseOverview). Twenty-one systematic reviews reporting on vitamin D supplementation in children from 2008 to 2013 were evaluated and all population, dosing, outcome and methodology characteristics reported in more than 1/3 of these systematic reviews were included in the online database (Dataset S1).

## Validity, utility and accessibility

Comprehensiveness of the database was evaluated using the search results from 13 systematic reviews (Table 5) not included in the original literature search (*McNally et al., 2015*). To be included in the validation, the systematic review had to: (1) assess trials of vitamin D supplementation; and (2) contain at least one prospective pediatric trial. Systematic reviews were excluded if they were published by one of our authors, or the reference list was screened as part of the literature search for our previous systematic review (*McNally et al., 2015*). Validation was performed by an independent author (LZ), who was not involved in the development of the database, and was blinded to the search results of the individual systematic reviews. This individual was provided with the eligibility criteria for each systematic review and then used the online database to identify the list of trials that would need further screening. Trial eligibility was further confirmed by a second author (NN). Sensitivity of the database was determined by comparing the number of trials in the database to the number identified within the individual systematic reviews (gold standard). The utility of the database was assessed by determining whether application of the database would have reduced the number of abstracts and full text articles for review. This was performed by a blinded author and using the population, age and outcome filters. Finally, accessibility was assessed using the Minervation validation instrument for healthcare websites (LIDA tool). This validation tool has been previously validated and used in several fields (*Nankervis, Maplethorpe & Williams, 2011*; *Pithon & Santos dos, 2014*; *Carlsson et al., 2015*; *Redmond et al., 2015*; *Küçükdurmaz et al., 2015*). Accessibility is addressed by looking at page setup, access restrictions, amount of outdated code and compatibility with NHS directives.

# RESULTS

## Search results

Figure 1 demonstrates the flow of studies identified by the search strategy. In total, 2,304 unique records were retrieved from the original electronic search with an additional 146 citations found in the reference lists of systematic reviews and eligible articles. Updating the search in January 2015 added an additional 129 records. Of the 2,579 articles, 2,188 were excluded at level one, with an additional 135 excluded at level two screening. In total, we identified 256 publications that reported on the results of a clinical trial administering any dose of ergocalciferol or cholecalciferol to children. From these 256 articles, 169 articles met eligibility criteria (Appendix S3). Further inspection identified 6 instances of double publications of the same trial with reporting of nearly identical findings (Table S3). These duplicate publications were not included in the study, resulting in a total of 163 publications.

## Study design, participant number and quality

The 163 publications evaluated 181 distinct study populations, included 365 separate arms, and enrolled a total of 18,539 children (Fig. S1). As shown in Table 1, RCTs contributed to the majority of the trials ($n = 108/163$, 66%) and patients ($n = 15,728$, 84.8%). Assessment of trial quality determined that 23% ($n = 38/163$) and 42% ($n = 69/163$) were at low

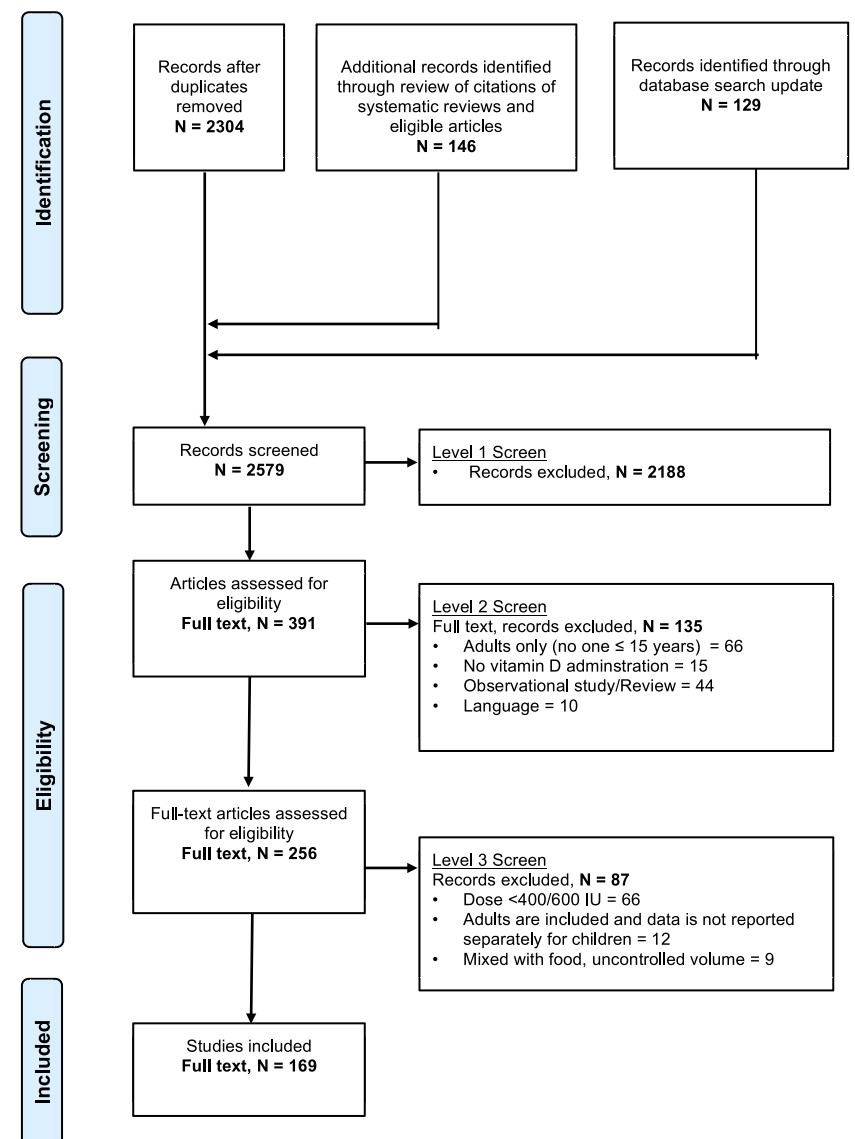

**Figure 1** **Flow chart of study selection based on inclusion and exclusion criteria.** The stages of a systematic selection scheme include: identification, screening, eligibility, and final included studies.

or medium/uncertain risk for bias, respectively. Of the 365 study arms, 263 (72.1%) administered one or more doses of vitamin D meeting our eligibility criteria, on a total of 11,947 children. The median number of participants in the high dose arms was 25 (IQR: 14–42). The 163 trials were published over a 46-year period between 1969 and 2014 (inclusive). The rate of trial publication changed significantly over time ($p < 0.001$), increasing from 1 trial per decade (1960–1969) to 15 trials in 2014 alone (Fig. 2A). Of the 163 trials, almost half ($n = 72$, 44%) have been published in the last 5 years (2010–2014). Compared to a linear model, the change over time better fits an exponential function with the number of trials doubling every 12.7 years ($R^2 = 0.85$ vs. 0.96 respectively).

**Table 1** **Assessment of study design and methodological quality.** Trials enrolled a total of 18,539 patients and a median (IQR) of 49 (25–94). High dose arms enrolled 11,947 patients with a median of 25 (14–42).

| Study characteristic | Trials/populations | %[a] |
|---|---|---|
| **Study design**[b] | | |
| RCT/qRCT | 108 | 66 |
| Single arm | 42 | 26 |
| Controlled, other | 13 | 8 |
| **Randomized trial quality**[b,c] | | |
| Low risk | 38 | 23 |
| Medium risk/unclear | 69 | 42 |
| High risk | 56 | 34 |
| **Cochrane risk of bias**[b,c,d] | | |
| Generation adequate | 56/57 | 34/35 |
| Concealment adequate | 50/62 | 31/38 |
| Blinding adequate | 47/25 | 29/15 |
| Outcome report complete | 124/36 | 76/22 |
| Outcome not selective | 89/6 | 55/4 |
| **Age groups**[e] | | |
| Neonates | 58 | 32 |
| Infants | 35 | 19 |
| Toddlers | 61 | 34 |
| Schoole rs | 104 | 57 |
| Adolescents | 82 | 45 |

**Notes.**

Abbreviations: (q)RCT, (Quasi randomized controlled trial).

[a]Because of rounding, percentages may not total 100.

[b]Values represent the number of trials, and the percentage out of the 163 identified trials.

[c]Studies were assessed using Cochrane risk of bias tool (*Higgins & Green, 2011*).

[d]For the Cochrane assessment, the number of trials where the risk of bias was unclear, we indicated their numbers after the '/'.

[e]Numbers of populations in each age group out of 181 populations. Numbers will add up to more than 181 populations as some included children from two or more age groups.

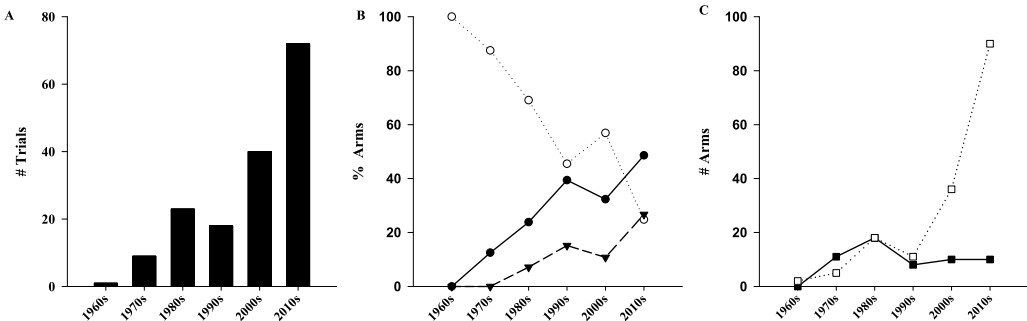

**Figure 2** **Evolution of pediatrics trials of high dose vitamin D over time.** Exponential increase in number of trials (A, $R^2 = 0.96$, $p < 0.001$). (B) Comparison of studied populations among the different decades ($p < 0.001$). (C) Comparison of form of vitamin D administered among different decades ($p < 0.001$). (●) Healthy/subclinical VDD; (○) Classical; (▼) Non-classical; (□) Cholecalciferol; (■) Ergocalciferol.

**Table 2 Diagnostic categories and outcomes of studied populations.** A total of 263 arms were identified as high dose. Populations in these arms were classified as conditions classically or non-classically associated with vitamin D deficiency. Details provided in (Table S4). Last column identifies the number of arms that administered high-dose vitamin D to 50 patients or more, and that were determined to be at a low-risk of bias.

| Diagnostic category | Arms | Patients | # Trials of low-risk of bias recruiting ≥50 patients |
|---|---|---|---|
| **Conventional outcomes** | | | |
| **Healthy/subclinical VDD** | **97** | **4,608** | **11** |
| **Classical diseases** | **123** | **4,134** | **1** |
| Premature/low birth weight | 48 | 2,127 | 1 |
| Rickets | 43 | 1,359 | 0 |
| Malabsorption | 15 | 319 | 0 |
| Epilepsy/seizure | 7 | 125 | 0 |
| Renal disease | 4 | 96 | 0 |
| Other | 6 | 108 | 0 |
| **Non-conventional outcomes** | | | |
| **Non-classical diseases** | **43** | **3,205** | **7** |
| Obesity | 7 | 213 | 0 |
| Asthma | 4 | 101 | 1 |
| Pneumonia/URTI | 4 | 2,065 | 4 |
| Recurrent acute otitis media | 3 | 251 | 1 |
| HIV | 3 | 65 | 0 |
| Dental fluorosis | 3 | 55 | 0 |
| Other | 19 | 455 | 1[a] |

**Notes.**

Abbreviations: URTI, Upper respiratory tract infections; VDD, vitamin D eficiency.
[a]Tuberculosis.

## Populations

The number and percentage of study arms recruiting populations within five age categories is provided in Table 1. Specific age categories were targeted in 61 arms, with the majority of those focusing on neonates ($n = 47/61$, 77.0%). Study arms involving populations with classic vitamin D related diseases were the largest category ($n = 123/263$, 46.8%), with prematurity ($n = 48$, 18.3%) and rickets ($n = 43$, 16.3%) representing the most common subpopulation (Table 2). Of the remaining 140 high-dose arms, 97 recruited healthy patients and focused on a classical role of vitamin D (25OHD or bone health) and/or prevention of rickets. The least common category, representing 16.3% of arms ($n = 43/263$) were those with diseases or outcomes less classically related to vitamin D. Of these, 12 included healthy patients and focused on primary prevention of non-classical conditions (respiratory infections, diabetes), and the remaining 31 enrolled participants with wide range of non-classical illness at baseline (Table S4), and administered vitamin D as a sole or part of the treatment plan. The proportion of arms evaluating non-classical populations or outcomes has increased significantly ($p < 0.001$), rising to 26.7% during the current decade (Fig. 2B). The number of low risk of bias arms enrolling more than 50 participants is presented in Table 2.

**Table 3 Characteristics of vitamin D supplementation in the 263 high dose study arms.** Vitamin D dosing regimens were placed into one of three frequency groups (daily, weekly/biweekly, and single/intermittent). Variable dosing regimens administered doses that are dependent on weight, age or body surface area (BSA).

| Supplementation | Arms | % [a] |
|---|---|---|
| **Dosing regimen** | | |
| Constant | 224 | 85.2 |
| Variable | 39 | 14.8 |
| **Dosing groups** | | |
| RDA/AI–999 | 50 | 19.0 |
| 1,000–3,999 | 73 | 27.8 |
| 4,000–39,999 | 33 | 12.6 |
| ≥40,000 | 107 | 40.7 |
| **Route[b]** | | |
| PO | 238 | 90.5 |
| IM/IV | 24 | 9.1 |
| **Form[c]** | | |
| D3 | 162 | 61.6 |
| D2 | 57 | 21.7 |
| **Frequency** | | |
| Daily | 137 | 52.1 |
| Intermittent/single dose | 96 | 36.5 |
| Weekly/biweekly | 30 | 11.4 |

Notes.

Abbreviations: AI, Adequate intake; BSA, body-surface-area; D2, ergocalciferol; D3, cholecalciferol; PO, enteral dosing; IM, intramuscular; IV, intravenous; RDI, recommended daily intake.

[a] Because of rounding, percentages may not total 100.

[b] In 1 arm, the route was unclear.

[c] In 44 cases vitamin D form was nuclear.

## Dosing regimens

Evaluation of dosing regimen characteristics identified the main supplement and route as cholecalciferol ($n = 162/263$, 61.6%) and enteral ($n = 238/263$, 90.5%), respectively (Table 3). Regarding supplementation frequency, 137 arms (52.1%) delivered drug on a daily schedule, and 96 (36.5%) used intermittent loading therapy. Most of the arms used a constant dose of vitamin D ($n = 224/263$, 85.2%), while the remaining used dosing based on age/weight or body surface area ($n = 27/263$, 10.3%), baseline 25OHD ($n = 7/263$, 2.7%), or initial response to supplementation ($n = 5/263$, 1.9%). Doses higher than 40,000 IU were the most common dose group, being used in 107 arms (40.7%). Whether dosing regimen characteristics changed over time was further investigated (Fig. 2C); ergocalciferol (D2) was more common prior to the 1980s, with cholecalciferol (D3) gaining significant attention over the past two decades ($p < 0.001$). In the present decade, D3 and D2 were used in ($n = 90/105$, 86%) and ($n = 10/105$, 10%) of the high-dose arms respectively (remainder unclear). No other change over time was evident for frequency of administration, choice of variable dosing, or route of administration (Fig. S2).
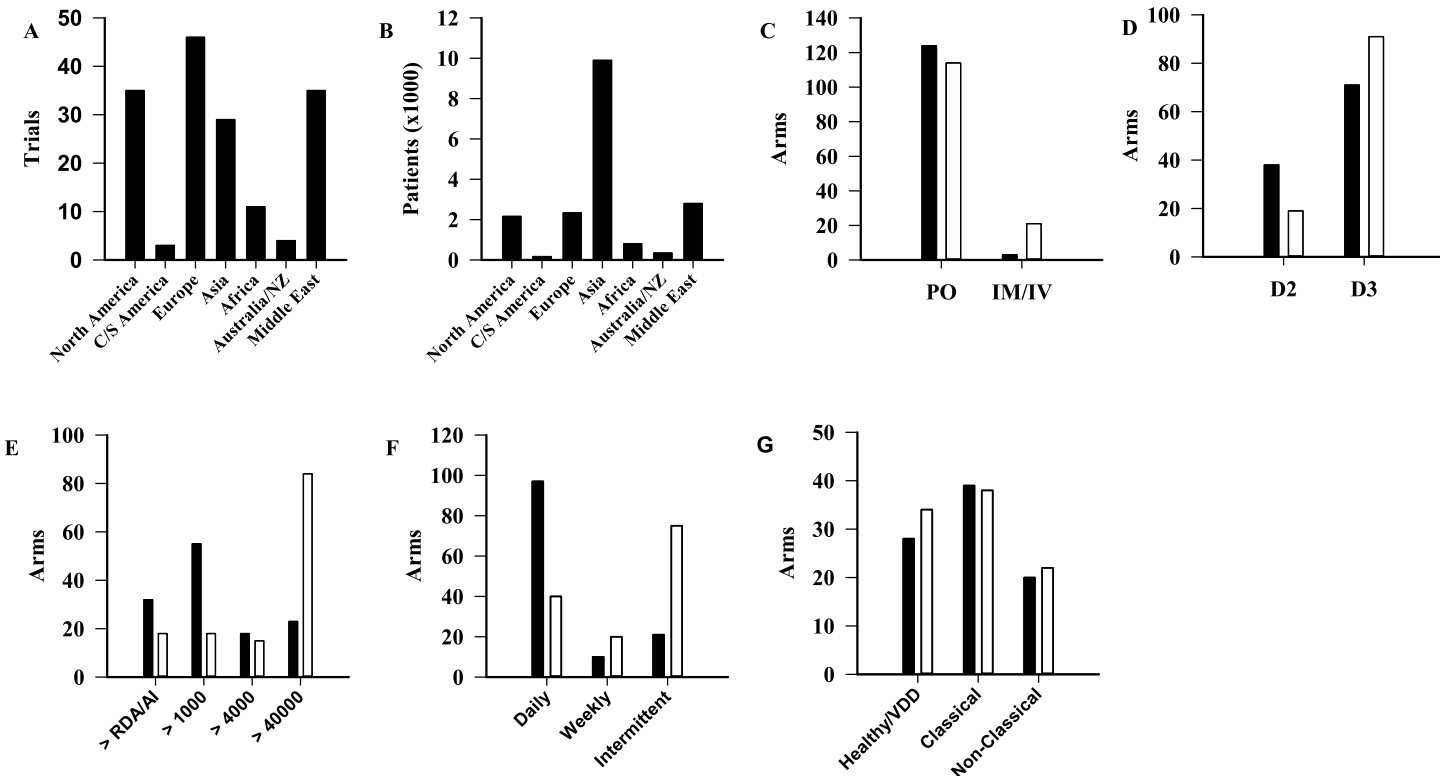

**Figure 3 Comparison of trials among geographical regions.** Number of published trials per region (A, $p < 0.001$), and patients (B, $p < 0.001$). (C–F) North America and Europe (■) compared to the other regions (□), in terms of route (C, $p < 0.001$), form (D, $p = 0.003$), dosage (E, $p < 0.001$), frequency of supplementation (F, $p < 0.001$), and population (G, $p = 0.81$). AI, Adequate Intake; C/S, Central/southern; D2, Ergocalciferol; D3, Cholecalciferol; IM, Intramuscular; IV, Intravenous; NZ, New Zealand; PO, Oral; RDA, Recommended Dietary Allowance.

## Geographical regions

Trials were categorized by geographical region (Fig. 3A) ($p < 0.001$). The area with the most published trials was Europe ($n = 46/163$, 28%) with North America and the Middle East each contributing 35 trials (21%). For comparison, the number of total participants by geographical region is shown in Fig. 3B ($p < 0.001$). Difference in dosing regimen preference by geographical region was also evident. Doses higher than 40,000 IU were sparse in Europe ($n = 14/263$, 5.3%) and North America ($n = 9/263$, 3.4%), but represented the most commonly used regimen in every other region (Fig. S3). In addition to dose, statistically significant differences were observed between North America and Europe when compared to the rest of the world for route, form and dosing frequency (Fig. 3). A comparison of the population types studied in these two continents, with the remainder of the world, did not identify statistically significant differences ($p = 0.81$) (Fig. 3G).

## Outcomes

Primary outcome varied among trials, with 106 (65%) targeting a biochemical marker, and 62 (38%) focusing on a clinical outcome (Table 4). Considering all outcomes reported, 25OHD level was the most common, being studied in 133 trials (82%), with blood calcium

**Table 4  Classification of studied outcomes of the 163 identified trials.**

| Outcome | Trials | %[a] |
|---|---|---|
| **Primary outcome**[b] | | |
| **Biochemical** | 106 | 65 |
| **Clinical** | 62 | 38 |
| Rickets/bone muscle mass | 35 | 21 |
| Non-classical clinical outcomes | 27 | 17 |
| **Total outcomes** | | |
| **Biochemical** | 156 | 96 |
| 25OHD | 133 | 82 |
| Blood calcium | 118 | 72 |
| Phosphate | 80 | 49 |
| PTH | 69 | 42 |
| ALP | 65 | 40 |
| Urine calcium | 41 | 25 |
| 1,25-(OH)2-D | 30 | 18 |
| Calcium absorption | 7 | 4 |
| **Clinical** | 99 | 61 |
| Bone mass | 47 | 29 |
| Rickets | 33 | 20 |
| Immuno-inflammatory | 19 | 12 |
| Respiratory | 9 | 6 |
| Cardiovascular | 8 | 5 |
| Renal | 6 | 4 |
| Diabetic | 6 | 4 |
| Hematological | 4 | 2 |
| **Other** | | |
| Anthropological measures | 20 | 12 |
| Adverse effects | 13 | 8 |

Notes.

Abbreviations: ALP, Alkaline phosphates; PTH, parathyroid hormone.

[a] Because of rounding, percentages may not total 100.

[b] Primary outcomes count exceeds 163, as 5 trials had both clinical and biochemical primary outcomes.

($n = 118/163$, 72%), phosphate ($n = 80/163$, 49%) and PTH ($n = 69/163$, 42%) being the next most common. For studies focused on non-classical diseases, outcomes evaluating the immunologic, respiratory and cardiovascular systems were studied in 19 (12%), 9 (6%) and 8 (5%) trials.

## Database validation

The thirteen systematic reviews identified 38 trials meeting our high-dose criteria, 36 (94.7%) of which were contained within the online searchable database. The database identified an additional 16 trials that satisfied the inclusion criteria of one or more of these systematic reviews, and were published prior to the literature search. The two eligible trials not present within the database were determined to be not available as a full text (abstract) or the article could not be located ($n = 1$, the reference did not exist on any of the searched
**Table 5  Validation of the online database using 13 systematic reviews.** Comprehensiveness of the database was evaluated using the search results from 13 systematic reviews not included in the original literature search (2008–2015).

| Review | Trials in the review[a] | Eligible trials[b] | Trials on ODB[c] | RCTs missing | Sensitivity | Additional trials on ODB |
|---|---|---|---|---|---|---|
| 1. Bacchetta, J 2008 | 3 | 0 | 0 | 3: Dose ≤ RDA | 0/0 (NA) | 0 |
| 2. Das, JK 2013 | 5 | 1 | 2 | 4: Dose ≤ RDA | 1/1 (100%) | 1 (Vervel, 1997) |
| 3. Das, RR 2013 | 2 | 2 | 2 | 0 | 2/2 (100%) | 0 |
| 4. Fares, MM 2015 | 4 | 2 | 2 | 2: Dose ≤ RDA | 2/2 (100%) | 0 |
| 5. Zittermann, A 2014 | 13 | 10 | 15 | 3: Dose ≤ RDA | 10/10 (100%) | 5 (Guillemant, 1998; El-Hajj, 2006; Dahifar, 2007; Arabi, 2009; Majak, 2009) |
| 6. Riverin, BD 2015 | 8 | 6 | 5 | 1 Reference NA (Darabi, 2013) 2: Dose ≤ RDA | 5/6 (83.3%) | 0 |
| 7. Ali, SR 2015 | 5 | 3 | 3 | 2: Dose ≤ RDA | 3/3 (100%) | 0 |
| 8. Kerley, CP 2015 | 7 | 6 | 5 | 1: Dose ≤ RDA 1 Full text NA (Utz 1976) | 5/6 (83.3%) | 0 |
| 9. Hoffmann, MR 2015 | 1 | 1 | 1 | 0 | 1/1 (100%) | 0 |
| 10. Jamka, M 2015 (Sci Rep.) | 1 | 1 | 5 | 0 | 1/1 (100%) | 4 (Ashraf, 2011; Kelishadi, 2014; Poomthavorn, 2014; Nader, 2014) |
| 11. Jamka, M 2015 (Eur J Nutr.) | 2 | 2 | 3 | 0 | 2/2 (100%) | 1 (Belenchia, 2013) |
| 12. Zittermann, A 2015 | 4 | 3 | 7 | 1: Dose ≤ RDA | 3/3 (100%) | 4 (Morcos, 1998; El-Hajj, 2006; Arabi, 2009; Lewis, 2015) |
| 13. Jiang, W 2015 | 1 | 1 | 1 | 0 | 1/1 (100%) | 0 |
| **Total** | **56** | **38** | **51** | **18: Dose ≤ RDA 1 Full text NA 1 Reference NA** | **36/38 (94.7%)** | **15** |

**Notes.**

Abbreviations: ODB, Online database (http://www.cheori.org/en/pedvitaminddatabaseOverview).

[a]Counting only perspective trials that fell under the pediatrics range and that administered vitamin D.

[b]Trials that satisfied our inclusion criteria of being controlled prospective trial, administering a high dose of vitamin D to children.

[c]Counting only those published prior to the search dates of the systematic reviews.

databases or the journal repertoire). The 13 systematic reviews also included an additional 18 trials that provided supplementation with lower doses of vitamin D (≤RDA), and all were present in the list of 256 pediatric clinical trials identified as part of the stage 2 screening. A summary of the utility assessment is shown in Table S5. The literature search of the 4 full pediatric systematic reviews identified between 684 and 1,343 unique records for screening and between 21 and 274 articles for full text review. In comparison, the online database search by the blinded author and using the population, age and outcome filters yielded between 2 and 10 articles for full text review (no eligible studies were missed). The reduction in number of papers for full assessment was reduced by 85.2% (SD 13.4%).

The accessibility assessment was performed using an online component evaluating page setup, access restrictions, amount of outdated code and compatibility. On this, our online database scored 45/54 (83%) in terms of accessibility (Table S6).

## DISCUSSION

This systematic review sought to identify all published pediatric clinical trials of high dose vitamin D supplementation, and determined this to be a large and rapidly expanding area of clinical research. Descriptive analysis identified heterogeneity in important trial design characteristics, including a recent significant increase in studies evaluating a wide range of populations and outcomes not classically related to vitamin D. A comprehensive searchable online database was developed to aid clinicians and researchers in the identification and evaluation of trials relevant to their patient population or area of interest.

Our literature search identified 256 pediatric clinical trials of vitamin D supplementation, of which 169 included one or more study arms meeting our definition of high dose. Evaluation of publications over time further demonstrated high-dose vitamin D supplementation to be a rapidly expanding area of clinical research. With the exception of the 1990s, where a brief decline in publications was observed, the trial number has almost doubled each decade. The decline in publications on high-dose vitamin D may relate to a late 1980's publication reporting high hypercalcemia rates (34%) in young infants receiving 600,000 IU (*Markestad et al., 1987*). A detailed comparison of the change in publication rate with other areas in pediatric research was limited by the lack of well-done systematic reviews of similar design and scope. However, one excellent comparator is the recent review published by Duffett and colleagues (*2013*), wherein they demonstrated a constant linear rise in clinical trials in the pediatric critical care setting. Taken together, the results suggest that pediatric clinical trial literature is steadily increasing. The exponential growth of the randomized controlled trial literature has been previously demonstrated (*Tsay & Yang, 2005*; *Bastian, Glasziou & Chalmers, 2010*). Part of the value of systematic reviews is in summarizing the literature, as the rate of publication of RCTs makes it impractical for clinicians to keep up with the primary publications.

Further comparison of publication rates between these two pediatric studies suggested a faster rate of rise in trials on high dose vitamin D. For perspective, if the current rate is maintained there will have been more trials published between 2010 and 2019 then in the preceding 5 decades combined. Evaluation of study characteristics, including the change over time, provided some insight into why the publication rate may be rising faster than other areas. Importantly, this evaluation demonstrated that an increasing number of trials are focusing on populations or outcomes not classically related to vitamin D (*Canadian Agency for Drugs and Technologies in Health, 2015*). This shift is consistent with the substantial growth in observational literature over the past two decades linking vitamin D to a widespread number of disorders involving the immune, neurological, respiratory, and cardiovascular systems (*Brehm et al., 2010*; *Levin et al., 2011*; *Gray et al., 2012*; *McNally et al., 2012*; *Abrams, Coss-Bu & Tiosano, 2013*; *Cadario et al., 2015*). The decision to pursue clinical trials of high dose supplementation in these populations may relate to postulations made that higher 25OHD levels, relative to those achieved with RDA, may be required to achieve maximal benefit for non-musculoskeletal outcomes (*Hathcock et al., 2007*). Further, available literature also suggests that when compared to healthy patients, those

with acute and chronic disease may have a blunted response to usual doses of vitamin D (*McNally et al., 2015*).

As part of the descriptive analysis we also sought out heterogeneity in other trial design features, including dosing regimen characteristics. The main dosing regimen characteristics where uniformity was evident include use of the oral route for drug administration and choice of the cholecalciferol form. Consistency in these areas is expected, given that cholecalciferol has been suggested to have favorable metabolism and greater biological activity, and 25OHD is well accepted as the best biological marker of vitamin D status (*Melamed & Kumar, 2010*). Outside of route and form, significant heterogeneity in regimen dose and frequency was evident. Further exploration determined that some of the heterogeneity could be explained by the geographical origin of the trial. In North America and Europe, daily supplementation was by far the most common regimen, while the remainder of the world predominantly used single or divided loading doses well in excess of the IOM Daily Upper Tolerable Intake Level. These differences might explain why, in contrast to countries like Australia and New Zealand, guidelines originating out of North America do not offer any strategies based on weekly or less frequent loading doses (*Munns et al., 2006*; *Godel, 2007*; *Manaseki-Holland et al., 2010*). Similar to the shift from ergocalciferol to cholecalciferol over time, a study characteristic anticipated to be changing was the proportion of dosing regimens incorporating patient age or weight into dose selection (*Aguirre Castaneda et al., 2012*; *McNally et al., 2015*). Despite being well recognized in other areas of pediatric research, the need for age- or weight-based vitamin D dosing has only recently been acknowledged by agencies such as the IOM. Our analysis demonstrated that 90% of trials used a constant dose, with no evidence of a recent shift towards age or weight based practise.

Our study findings strongly suggest that identification and synthesis of the clinical trial literature on high dose vitamin D supplementation has been and will continue to be a challenge for clinicians, researchers and policy makers. First, we demonstrated that there is a significant and rapidly expanding body of clinical trial literature. Second, our descriptive analysis identified significant heterogeneity in multiple relevant study design characteristics, including population (age, disease), dosing regimen, and outcome selection. As many of these characteristics are often poorly described in titles and abstracts, end-users may struggle to not only locate relevant citations but also to determine whether the study will address the question(s). A solution to these two problems was provided as part of this study: a comprehensive online open access searchable database of pediatric clinical trials of high dose vitamin D, similar to what has been generated for other areas, including Eczema (*Nankervis, Maplethorpe & Williams, 2011*; *Nankervis et al., 2015*). Further, we completed a validation study proving the database to have excellent sensitivity, containing 36 of 38 of the pediatric high dose trials reported within 13 published independent systematic reviews. The validation work also identified that the database contained an additional 18 trials that met one or more of the systematic review eligibility criteria. This last finding suggests that researchers performing systematic reviews of vitamin D supplementation may already be struggling with the volume of literature and poor description of trials in titles, abstracts and keywords. In addition to being comprehensive, the database was further designed to
assist the end user with heterogeneity. Not only was the database designed to be searchable by key study characteristics, but also the outcome page was set-up to present the user with the study design, population, dosing regimen, and outcome data required to evaluate trial relevance. Utility of the database was also evaluated, by comparing with 4 pediatric systematic reviews, and when combined with the search functions, it was determined that the number of full text articles requiring review could have been reduced by 85% (*Das et al., 2013*; *Fares et al., 2015*; *Riverin, Maguire & Li, 2015*; *Ali & McDevitt, 2015*). These observations, combined with the fact that the database would reduce the time associated with developing and performing a literature search, indicate the database to be a beneficial resource for the field.

Finally, the availability of a comprehensive validated database allowed us to identify those areas where there may be sufficient evidence to answer questions regarding the clinical benefits of high-dose vitamin D. Considering all low-risk of bias studies, regardless of size, there were only two areas (respiratory infection/asthma, $n = 2,166$ and prematurity/low birth weight, $n = 2,127$) with more than 100 total children enrolled in the high-dose arms. Compared to reviews of other areas of pediatric research, the average number of children recruited per study was smaller (*Hamm et al., 2010*). The reason for this is unclear, but given the recent interest in the non-classical roles of vitamin D, much of the identified work may represent pilot work intended to precede large phase III studies (*Randolph & Lacroix, 2002*; *Nicholson et al., 2003*). Of the two areas with a relative breadth of evidence, pediatric respiratory illness and asthma have been recognized by research groups, culminating in multiple systematic reviews (*Charan et al., 2012*; *Das et al., 2013*; *Pojsupap et al., 2015*). Due to heterogeneity in population, dosing regimen and outcome characteristics these reviews were suggestive, but not definitive, for benefit and further research is underway. In contrast to the area of pediatric respiratory illness, there have not been any recent attempts to systematically synthesize the significant clinical trial literature in the area of prematurity and/or low birth weight. A systematic review of the effectiveness and safety in this population may be worthwhile as metabolic bone disease remains a problem and recent observational data suggests that vitamin D deficiency may augment non-musculoskeletal pathophysiology (*Onwuneme et al., 2012*). This work would benefit nutrition guidelines in NICU and inform dosing regimens for future phase III studies. Finally, it is important to note that with somewhere between one and two dozen new publications per year the areas with sufficient high-quality evidence to address clinical efficacy could change quickly.

Although this review has many strengths, a number of important limitations should be acknowledged. First, for the majority of the trials information was not available on potentially relevant study characteristics including race, UV exposure, diet, drug compliance and blood collection techniques. Second, the large volume of studies and significant heterogeneity in relevant study characteristics presented considerable challenges for synthesis and presentation of the studies in a manner useful to all clinicians and researchers. As has been successfully performed in other clinical areas, including eczema and pediatric intensive care, we sought to address this problem through the creation of a comprehensive accessible online searchable database of identified trials (*Nankervis, Maplethorpe & Williams, 2011*; *Duffett et al., 2013*). Using this database readers

and end-users should be able to quickly locate and evaluate the specific population, dosing regimen, outcome and/or study design features of interest. The searchable database also helps address another important study limitation created by the lack of an accepted definition or algorithm for high dose vitamin D. Recognizing the challenges involved with developing such a definition, we chose to use a very inclusive or sensitive threshold. We recognize that the true definition of high dose also incorporates dosing and patient factors including age, disease status, and dose frequency and duration. We have carefully designed the database to ensure that users have the ability to rapidly evaluate these characteristics and apply their own thresholds or algorithm for defining high dose. Finally, a significant limitation of the database is that it is at risk for going out of date. Our goal is to update this list yearly, and provide a data entry page for research groups to provide relevant information on their study.

## CONCLUSION

This systematic review identified 169 publications reporting on pediatric trials of high-dose vitamin D, and made relevant trial information available as part of an online searchable database. Importantly, this field has seen an increase in published trials over the past decade, spanning a wide range of populations, dosing regimens and outcomes. To assist the field we developed, validated and demonstrated utility of an online database of pediatric clinical trials of high dose supplementation vitamin D development. The availability of this database, combined with search functions and extracted data, should aid clinicians, researchers and policy makers. Using this resource, clinicians will be able to quickly and comprehensively identify and evaluate the level of clinical trial evidence for a particular patient population. Finally, the availability of an up-to-date list of published trials, combined with extracted information on population, eligibility criteria, dosing regimen and outcomes should expedite the systematic review process for researchers and policy makers.

### Funding

Nassr Nama received an Undergraduate Research Opportunity Program (UROP) from the University of Ottawa. Nassr Nama and Klevis Iliriani received Summer Studentships from the Children's Hospital of Eastern Ontario Research Institute. The funders had no role in study design, data collection and analysis, decision to publish, or preparation of the manuscript.

### Grant Disclosures

The following grant information was disclosed by the authors:
University of Ottawa.
The Children's Hospital of Eastern Ontario Research Institute.

### Competing Interests

Margaret Sampson is an Academic Editor for PeerJ.

## Author Contributions

- Nassr Nama and James D. McNally conceived and designed the experiments, performed the experiments, analyzed the data, contributed reagents/materials/analysis tools, wrote the paper, prepared figures and/or tables, reviewed drafts of the paper.
- Kusum Menon, Lauralyn McIntyre and Dean Fergusson wrote the paper, reviewed drafts of the paper.
- Klevis Iliriani, Supichaya Pojsupap and Katie O'Hearn performed the experiments, contributed reagents/materials/analysis tools, wrote the paper, reviewed drafts of the paper.
- Margaret Sampson conceived and designed the experiments, performed the experiments, contributed reagents/materials/analysis tools, wrote the paper, reviewed drafts of the paper.
- Linghong (Linda) Zhou performed the experiments, wrote the paper, reviewed drafts of the paper.

## Data Availability

The dataset is provided in the Supplemental Information.

## Supplemental Information

Supplemental information for this article can be found online at http://dx.doi.org/10.7717/peerj.1701#supplemental-information.

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
