# Peer review of "A systematic review of pediatric clinical trials of high dose vitamin D"

_PeerJ, doi:10.7717/peerj.1701_

## Round 0.1 · original submission · Major Revisions

I have a few minor additions to the reviewers' suggestions below:

Line 79: please reference the PRISMA guidelines
Line 101: spell out "3"
Figure 2: Needs a more detailed description (e.g. What is the "Field" you are refering to? What is plotted?). State what comparisons the P values for 2C and 2D are based on. The Y axis label for 2D is missing "%"

Reviewer 1 ·

Basic reporting

Please see general comments for the authors

Experimental design

Please see general comments for the authors

Validity of the findings

Please see general comments for the authors

Additional comments

November 17, 2015
Paper Review

Title: A systematic review of pediatric clinical trials of high dose vitamin D: Many publications, little quality evidence

Summary: The stated objectives of the study are to identify and summarize the literature on pediatric clinical trials of high dose vitamin D; to determine areas of high quality evidence to evaluate the benefits of high dose vitamin D on clinically relevant outcomes; and to develop a resource that makes the literature immediately accessible to clinicians, researchers and policy makers. The authors conducted a systematic review of trials up to January 2015 in which doses higher than 400 IU (< 1 year) or 600 IU (≥1 year) were included. The manuscript did not include any efficacy assessments or meta-analyses, but did made referral to an online resource that summarizes identified pediatric clinical trials of vitamin D. The authors conclude that despite the large number of trials, few trials were able to provide answers on clinical efficacy.

Assessment: The manuscripts addressed a contemporary research issue in the light of debates on recommendations for vitamin D supplement doses. Strengths of this study include the thorough systematic review conducted. The methodology for identifying and classifying the studies was sound. The consideration of studies in other languages was also a strength of the study. The online summary of the retrieved pediatric clinical trials seems a very valuable resource for researchers interested in conducting meta-analyses in the field of vitamin D supplementations in pediatric populations.

Specific comments:
• It was surprising that the manuscript stopped after presenting descriptives of the clinical trials and no efficacy assessment or meta-analyses were included. It gave the dissatisfaction of watching two periods of a good hockey match and to come to realize that the third period will not be played. We have not seen this format before and are not sure how readers will appreciate this ‘half manuscript’.
• As it stands, the manuscript leaves the reader wondering what the importance of this descriptive work is? For instance, the reader may wonder this where the authors mention and illustrate that the numbers of studies being published are increasing. Does it matter whether this increase is linear or exponential? Throughout the Discussion section there is this “so what?” that begs to be answered after reading the descriptive.
• The authors should consider including a summary table of the 169 articles included in the review. And to include the citations.
• The title of the paper should be revised. The current title is inappropriate for several reasons including the use of the term “evidence”. It is the meta-analysis that would establish the evidence rather than a narrative and since there were no meta-analyses there is no evidence that arises from this manuscript. A proper title should align with the study objectives. This could be along the lines of the objective as stated in lines 68… “A description of methodologies and outcome measures of pediatric clinical trials of vitamin D.”
• There are references to “high dose” and “low dose” in the paper which needs clarification. To characterize dosage, there is the need for a combination of the quantity of the dosage and frequency of the dosage. For example, 4000 IU per day may be considered high, and 4000 IU per month considered low. Additionally, because this is among children (who very substantial in body weight), there is the need to further take into account body weight when speaking to high and low. As such, doses are ideally expressed at IU per day per kg body weight.
• The authors state that the online resource would be useful to researchers, clinicians and policy makers. Agreed that this is very useful to researchers who are at the point to conducting meta-analyses. But the use to clinicians and policy makers is unclear. Could the authors justify/clarify this?
• The manuscript is very well written with very few typos. But please check: Line 25: … a resource that makes… Line 41: Referral to ‘non-classical diseases’ in the abstract needs explanation/to be avoided.

Reviewer 2 ·

Basic reporting

Labels on figures and tables are sometimes not clear. For example:
In the figure legend for figure C, does “healthy/subclinical VDD” refer to populations while “classical” and “non-classical” refers to outcomes? Are these mutually exclusive? It seems that there could be trials in “healthy/subclinical VDD” populations that look at either classical or non-classical outcomes. There also appears to be an error in the scale for figure 2A (“18” comes between “23” and “40”).
Table 1: Randomised trial quality is listed as “low risk” etc. Does this mean “low risk of bias”? How was it assessed? The meaning of the footnote ‘c’ is unclear.
Table 2: Figure legend: Were trials divided by population or outcomes? A more informative label for the third column would be useful.
Table 3: What is the difference between “intermittent” and “single dose” (in brackets)? Are these the same thing?
Figure 3: There are multiple abbreviations used in the x-axis labels that are not defined in the figure legend.
Minor comments:
1. Abstract methods: Rather than listing only the search date (January 2015) in methods, it would be more informative to provide a range of dates with start and end dates for which literature was included in the search. It might also be useful for the reader to include the age range in the abstract methods (0-18 years).
2. Line 88: suggest adding ”were considered eligible in these populations”
3. Line 105: should read “where the full text was assessed”
4. Lines 106-108: Not completely clear from the way this is written whether only the largest report was included in the analysis or whether separate reports were combined in some way.
5. Line 175: “the majority focusing on neonates (n=47/263, 17.9%)”: suggest rewriting as 47 is not the majority of 263.
6. Lines 243-246: Meaning unclear, please revise.
7. Figure 1: Box titled “Level 2 Screen” lists exclusion criteria as “no one <=15 years” while the methods states children up to 18 years were included. Please clarify.

Experimental design

1. Vitamin D has been hypothesised to play a role in the prevention of some conditions as well as in the treatment of others. For some conditions such as asthma trials exist of vitamin D for both prevention (e.g. maternal supplementation trials to prevent childhood asthma) and treatment (e.g. giving vitamin D to asthmatic individuals with low vitamin D levels to test whether it improves asthma symptoms or prevents exacerbations). It was not clear to me whether this systematic review included both types of trials. It would be useful to state this in the methods and potentially to separate studies according to whether vitamin D was trialled as a prevention or treatment strategy.
2. Presenting results in terms of “number of arms” seemed an unusual way to present the data and was somewhat confusing, particularly when figures referred to “% arms”. Could the data be presented as number of trials containing at least one high dose arm?
3. Does the increased rate of vitamin trial publication mirror an overall increase in the rate of publication of all clinical trials in general, or is there a specific increase in publication of vitamin D trials over and above a general increase in publication of other types of clinical trials? These two possibilities have different implications.
4. Presenting both the increasing number of trials published per year and the increasing number of patients enrolled in trials per year as two separate figures seems somewhat redundant since one would assume that the number of patients enrolled in trials would increase as the number of trials increases. It might be more informative to analyse whether there has been a change in the number of participants per trial over the relevant time period.
5. Some of the areas assessed are likely to be inter-related. For example, trials which aim to correct vitamin D deficiency may be more likely to use higher doses of vitamin D, and these sorts of trials may be more common in regions where vitamin D deficiency is more common, for example due to cultural practices (veils), racial differences (including skin colour) or lower UV levels. Please discuss where/whether you think this is a potential explanation for your findings. For example, could this explain the finding that higher doses were used in trials conducted outside Europe and North America?

Validity of the findings

Conclusion: The conclusion of the abstract might be better phrased as "[clinical efficacy] of high-dose vitamin D" for clarity.

Additional comments

No comments

---

## Round 0.2 · accepted · Accept

The authors have adequately addressed all the points raised by the reviewers.

Reviewer 2 ·

Basic reporting

The authors have adequately addressed my previous comments.

Experimental design

The changes to the manuscript have improved the clarity of the aims and more clearly described how this work contributes to the field of research.

Validity of the findings

The addition of a section assessing validity of the database by comparing with published systematic reviews is very useful.

Additional comments

No additional comments